# Numerical Model for Calculating the Unstable State Temperature in Asphalt Pavement Structure

**Naiji Zhang [1], Guoxiong Wu [2], Bin Chen [3,4,\*] and Cong Cao [1]** 

[1]  College of Civil Engineering, Chongqing Jiaotong University, Chongqing 400074, China; zhangnaiji2003@163.com (N.Z.); cong_c1007@163.com (C.C.)

[2]  School of Material Science and Engineering, Chongqing Jianzhu College, Chongqing 400074, China; wuguoxiong@cqjtu.edu.cn

[3]  Department of Civil Engineering, Zhejiang University City College, Hangzhou 310015, China

[4]  Yangtze Delta Institute of Urban Infrastructure, Hangzhou 310015, China

\*  Correspondence: chenbin@zucc.edu.cn or jeetchen_123@hotmail.com

**Abstract:** In this study, we determined the factors that influence of the temperature on an asphalt pavement by developing a two-dimensional unsteady temperature numerical calculation model using the finite difference method and Matlab. Based on the temperatures obtained by a buried sensor in a construction project, we collected the temperatures at different depths in the pavement structure in real time, and we then compared and analyzed the calculated and measured data. The results showed that the temperature in the asphalt pavement structure was significantly correlated with meteorological factors, such as the air temperature, but it also exhibited obvious hysteresis. Compared with the measured data, the maximum deviation in the numerical model based on the variations in the atmospheric temperature and solar radiation was 3 °C. Thus, it is necessary to effectively optimize the selection of asphalt pavement materials by simulating the temperature conditions in the asphalt pavement structure.

**Keywords:** asphalt pavement; finite difference; numerical calculation model; unstable state temperature

## 1. Introduction

Asphalt is a typical temperature-sensitive material, and thus its mechanical properties and the performance of asphalt pavements vary significantly with the temperature [1]. Continuous periodic changes in environmental factors make the pavement structure produce an unstable heat flow, so the temperature distribution inside the pavement structure is not uniform and stable, where this process is complex and changeable in both time and space [2].

Previous studies have investigated the temperature field in the layers of asphalt pavement structures. However, the theoretical deduction of the heat conduction formula includes many assumptions and some parameters are difficult to obtain, so it has limited practical applications [3]. Statistical formulae based on measured data require large amounts of field data, which are discrete and they lead to problems regarding regional applicability, and thus they are rarely representative [4]. The BELLS3 model has been applied widely to asphalt temperature prediction in the United States, but is not ideal in terms of its accuracy when applied to thick asphalt pavements [5]. Viljoen used a sine function and exponential function to fit the changes in temperature at different depths during the day and night. The fitting effect was better for the surface temperature on sunny days, but it cannot be applied to estimate the pavement temperature during cloudy weather and at different depths because the angular frequency of the sine function and the time of the lowest temperature in the day are fixed values [6].

In the present study, in order to solve the problem associated with numerous input parameters and low accuracy numerical calculations, we established a numerical calculation model of an asphalt pavement structure at different temperatures based on meteorological factors, such as the atmospheric temperature and solar radiation, according to the theory of heat transfer. We also conducted comparative analyses with measured data to determine whether the model could estimate the temperature in the asphalt pavement structure, thereby providing a theoretical basis for studying the structural behavior of asphalt pavements.

## 2. Factors that Influence the Temperature in Asphalt Pavement Structures

### 2.1. Meteorological Elements

An asphalt pavement is affected greatly by the natural environment, including constantly changing external environmental factors, such as solar radiation, external temperature, air flow, and rainfall, as shown in Figure 1. The three forms of heat and energy exchange that occur in the structure of an asphalt pavement comprise radiation, convection, and conduction. Some of the solar radiation and sky radiation are reflected by the road surface, and the rest is absorbed and transformed into heat energy. This part of the heat energy is superimposed on the external temperature, thereby resulting in a high surface temperature, which generates heat conduction mainly along the thickness of the road surface to a lower temperature region.

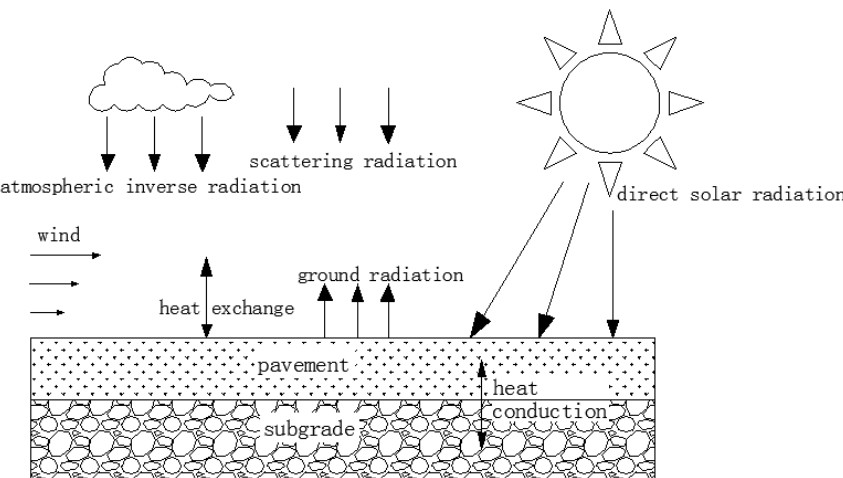

**Figure 1.** Schematic diagram illustrating the heat exchange process in asphalt pavement.

During the day, meteorological factors can change constantly, thereby resulting in unstable heat flow. Therefore, the temperature distribution inside the pavement structure is also uneven and unstable. The reasonable determination of these factors is highly significant for the accurate numerical calculation of the temperature field in asphalt pavements. A previous analysis showed that, among the many meteorological factors, solar radiation, the atmospheric temperature, and sunshine hours mainly affected the temperature of asphalt pavements. Therefore, these three factors were considered in a numerical calculation model for estimating the temperature in an asphalt pavement [7].

These meteorological data can be obtained from meteorological stations. However, these discrete data cannot be applied in calculations using numerical models. In a previous mathematical simulation of variations in meteorological elements, based on meteorological data obtained in Chongqing during 20 years from 1995 to 2015, tests of the simulation equation using meteorological elements showed that the results met the engineering requirements [8].

### 2.2. Comprehensive Heat Transfer Coefficient h

In the heat transfer process between an asphalt pavement and the outside world, there are two modes of heat transfer: convection heat transfer and radiation heat transfer. In engineering, the heat transfer coexisting convection and radiation is called the comprehensive heat transfer. The heat flux of the comprehensive heat transfer is calculated as follows:

$$q = q_c + q_r = (h_c + h_r)(t_w - t_a) = h(t_w - t_a) \tag{1}$$

The comprehensive heat transfer coefficient $h$ is the sum of the convective and radiative heat transfer coefficients:

$$h = h_c + h_r = 2.6\left(\sqrt[4]{t_w - t_a} + 1.54v\right) + \varepsilon \cdot C_0 \cdot \frac{\left[\left(\frac{T_w}{100}\right)^4 - \left(\frac{T_a}{100}\right)^4\right]}{t_w - t_a} \tag{2}$$

At present, there is no consensus understanding of the value of the comprehensive heat transfer coefficient $h$. Equation (2) shows that the value is closely related to the wind speed $v$, atmospheric temperature $t_a$, and surface temperature $t_w$, but the wind speed, temperature, and surface temperature can change constantly during the day, so the comprehensive heat transfer coefficient also changes with time. However, the numerical calculation will be very difficult if the comprehensive heat transfer coefficient is expressed as a function of time.

Equation (2) also shows that the influence of temperature on the convective heat transfer coefficient is limited, whereas the influence of the wind speed $v$ is dominant and the effect of changes in the radiative heat transfer coefficient is very small in most cases. Considering that the cooling process is often accompanied by a drastic change in wind speed, we propose to establish a function of the comprehensive heat transfer coefficient by using the wind speed as an independent variable, as follows:

$$h = 3.7073v + 9.4455 \tag{3}$$

The average wind speed $v$ during a day taken based on the average data provided by the weather station, ranges from 0 to 5 m/s.

### 2.3. Thermophysical Parameters of Materials

In addition to external meteorological factors, the thermophysical parameters of asphalt pavement materials affect the changes in temperature in an asphalt pavement. The thermophysical parameters of an asphalt pavement are related to its porosity, moisture content, and material composition, and they greatly influence the properties and gradation of materials [9]. The absorptivity of the road surface with respect to solar radiation has an important role in determining the temperature distribution in an asphalt pavement. A small change in the absorptivity value will also lead to a large change in the pavement temperature [10]. The solar radiation absorptivity of a road surface is closely related to the road surface conditions (surface wear, etc.) and solar elevation. When a road surface is smoother, a highly reflective surface will be readily formed and the radiation absorptivity of the road surface will be reduced.

Based on checks and comparisons, the recommended values for asphalt pavements are shown in Table 1 [11]. For a common semi-rigid base asphalt pavement, Table 2 shows the recommended values for the specific heat $c$, density $\rho$, thermal conductivity $\lambda$, and thermal diffusivity $\alpha$ in pavement materials [12].

**Table 1.** Solar radiation absorption rate.

| Pavement Condition | |
|---|---|
| General | Smooth |
| 0.86~0.88 | 0.82 |

**Table 2.** Thermophysical parameters.

| Materials | $c$, J/(kg·°C) | $\rho$, kg/m³ | $\lambda$, W/(m·°C) | $a$, m²/h |
|---|---|---|---|---|
| Coarse grain type | 900 | 2050 | 1.2~1.4 | 0.0028 |
| Medium grain type | 850 | 2100 | 0.9~1.2 | 0.0024 |
| Fine grain type | 800 | 2000 | 0.8~1.0 | 0.0020 |

## 3. Numerical Model for Calculating the Temperature in an Asphalt Pavement Structure

### 3.1. Numerical Model for Calculating the Temperature

An asphalt pavement is continuously affected by natural factors, such as the ambient air temperature, solar radiation, and wind speed. These natural factors change constantly during the day and the heat transfer in the asphalt pavement is also unstable, so the temperature distribution in an asphalt pavement can be treated as an unsteady thermal conduction problem. In addition, the length and width of an asphalt pavement are much larger than its thickness, although not by an order of magnitude, and thus it can be approximated as a large flat object [13]. After ignoring the effects of the solar altitude angle and other factors, the effects of external factors on the temperature of an asphalt pavement are consistent with its width and length. The temperature field in an asphalt pavement is mainly related to the transfer of temperature in the thickness direction. Therefore, the temperature distribution in an asphalt pavement structure can be treated as a linear heat conduction problem.

The temperature of an asphalt pavement structure is actually a two-dimensional unsteady heat conduction problem. Unlike steady-state heat conduction, the temperature of each node in an asphalt pavement changes according to the position, but also with time [14]. From an energy balance viewpoint, a grid element imports or exports heat between adjacent grid elements, but its own energy also changes with time [15]. Therefore, for the unsteady heat conduction problem, in addition to dividing the object into grid elements in space, time should be divided into many intervals, where the order of the time intervals is expressed in *K*. The solution of the unsteady heat conduction problem is based on the initial time $\tau = 0, \Delta\tau, 2\Delta\tau, \ldots, K\Delta\tau, \ldots$ The temperature values for each node in an object at different times are then obtained in turn. Using this approach and according to the basic principle of heat transfer, we established a numerical model for calculating the temperature in an asphalt pavement structure by using the finite difference method, as shown in Figure 2.

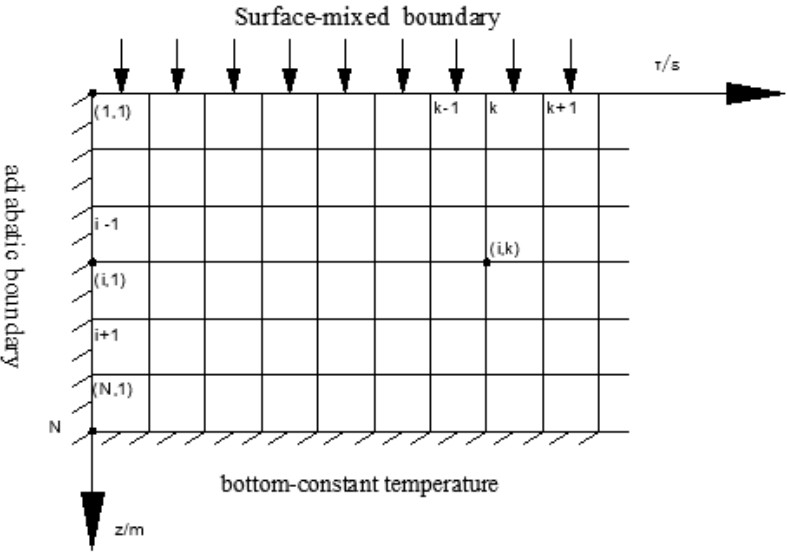

**Figure 2.** Model for predicting the temperature in an asphalt pavement.

Model description:

- The asphalt pavement structure can be regarded as having a finite thickness, so the pavement is divided into N layers according to the distance along the Z direction. The N layers represent the bottom of the base, which is the maximum thickness considered by the model. The time starts from $\tau = 0$ and is divided according to the division. $i$ denotes the location of the node, $k$ denotes the k moment, and $t_i^k$ denotes the temperature of node $i$ at time $k$.
- The asphalt surface is affected by many natural factors, such as the atmospheric temperature and radiation, so each factor has a different heat transfer effect. Therefore, in order to comprehensively consider the effects of various external factors, a mixed boundary is adopted on the surface of the asphalt surface.
- Considering that the transverse distance of the pavement structure is much larger than the depth, and that the effects of external factors on the transverse direction of the pavement structure are consistent, the heat transfer along the transverse direction of the asphalt pavement is neglected [16]. In the calculation model, the lateral direction is regarded as the adiabatic boundary.
- Studies have shown that the effects of external factors on the temperature of an asphalt pavement structure depend on the depth, and that a threshold exists beyond which the temperature of the asphalt pavement does not change greatly [17]. Therefore, the bottom of the base is regarded as a constant temperature boundary in the calculation model.

### 3.2. Nodal Difference Equation in the Model

The temperature distribution in an asphalt pavement can be regarded as a two-dimensional unsteady heat conduction problem with constant physical properties and no internal heat source. The differential Equation for heat conduction is as follows [18]:

$$\frac{\partial t}{\partial \tau} = a\frac{\partial^2 t}{\partial z^2}, \tag{4}$$

The node discrete equation is written for its internal nodes $P$ $(i, k)$. The second derivative of temperature with respect to $Z$ is expressed by the central difference:

$$\left(\frac{\partial^2 t}{\partial z^2}\right)_{i,k} = \frac{t_{i-1}^k - 2t_i^k + t_{i+1}^k}{\Delta z^2}, \tag{5}$$

The backward difference is used for the first derivative of temperature versus time:

$$\left(\frac{\partial t}{\partial \tau}\right)_{i,k} = \frac{t_i^k - t_i^{k-1}}{\Delta \tau}, \tag{6}$$

Equations (5) and (6) can be substituted into Equation (4) to obtain:

$$\left(1 + 2\frac{a\Delta\tau}{\Delta z^2}\right)t_i^{k+1} = \frac{a\Delta\tau}{\Delta z^2}(t_{i-1}^{k+1} + t_{i+1}^{k+1}) + t_i^k, \tag{7}$$

i.e.,

$$(1 + 2Fo)t_i^{k+1} = Fo(t_{i-1}^{k+1} + t_{i+1}^{k+1}) + t_i^k, \tag{8}$$

### 3.3. Model Boundary Node Difference Equation

Considering the combined heat transfer from natural factors such as temperature and solar radiation, the total heat transfer in an asphalt pavement surface is expressed as follows:

$$\Phi = h(t_a - t_w) \cdot A + \alpha \cdot I(\tau) \cdot A$$
$$= hA[(t_a + \tfrac{\alpha \cdot I(\tau)}{h}) - t_w] \tag{9}$$
$$= hA(t_e - t_w)$$

where $t_e = t_a + \frac{\alpha \cdot I(\tau)}{h}$ is the synthetic temperature, which is used to comprehensively consider the effects of the atmospheric temperature and solar radiation on the temperature distribution in an asphalt pavement.

Thus, for the heat flow density in an asphalt pavement surface $q(\tau) = \frac{\Phi}{A} = h(t_e - t_w)$, the mixed boundary conditions can be expressed as follows.

$$-\lambda \frac{\partial t}{\partial z}\Big|_{z=0} = q(\tau) = h(t_e - t_w), \tag{10}$$

For boundary nodes $(1, k)$, the implicit difference scheme based on the thermal balance method is as follows.

$$h(t_e^{k+1} - t_1^{k+1}) - \lambda \frac{t_1^{k+1} - t_2^{k+1}}{\Delta z} = \rho c \frac{t_1^{k+1} - t_1^k}{\Delta \tau} \cdot \frac{\Delta z}{2} \tag{11}$$

We then obtain:

$$t_2^{k+1} - t_1^{k+1} + \tfrac{h \cdot \Delta z}{\lambda}(t_e^{k+1} - t_1^{k+1}) = \tfrac{1}{2} \tfrac{\rho c \cdot \Delta z^2}{\lambda \cdot \Delta \tau}(t_1^{k+1} - t_1^k),$$
$$\tfrac{h \cdot \Delta z}{\lambda} = Bi, \; \tfrac{\lambda \cdot \Delta \tau}{\rho c \cdot \Delta z^2} = Fo, \tag{12}$$

Therefore:

$$t_2^{k+1} - t_1^{k+1} + Bi(t_e^{k+1} - t_1^{k+1}) = \frac{1}{2Fo}(t_1^{k+1} - t_1^k), \tag{13}$$

The implicit difference expressions for the mixed boundary nodes are obtained by shifting and sorting out the terms as follows.

$$(1 + 2Bi \cdot Fo + 2Fo)t_1^{k+1} = 2Fo(t_2^{k+1} + Bi \cdot t_e^{k+1}) + t_1^k, \tag{14}$$

After deduction, the difference Equation for each node in the numerical model for calculating the temperature in an asphalt pavement is shown in Table 3.

**Table 3.** Node differential equation in the model.

| | | |
|---|---|---|
| Internal Node | | $(1 + 2Fo)T_i^{k+1} = Fo(T_{i-1}^{k+1} + T_{i+1}^{k+1}) + T_i^k$ |
| Boundary Node | Mixed | $(1 + 2Bi \cdot Fo + 2Fo)T_1^{k+1} =$ $2Fo(T_2^{k+1} + Bi \cdot T_e^{k+1}) + T_1^k$ |
| | Constant | $T_N^i = T_0$ |

## 4. Analysis of Calculated Results

### 4.1. Case Study

Chongqing is located inland in southwestern China and in the upper reaches of the Yangtze River. Its landforms mainly comprise hilly and mountainous areas, where mountainous areas account for 76% of the total, and thus it is known as the "mountain city" Chongqing has a subtropical monsoon humid climate, with a hottest monthly average temperature of 28.3 °C, coldest monthly average

temperature of 7.8 °C, and annual average temperature of 18.4 °C. Banan District is located in the south of Chongqing's main urban area and in the hilly area on the south bank of the Yangtze River. The geological and geomorphological forms of Banan District are complex and diverse, with high topography in the south and low topography in the north, and large fluctuations.

In order to verify the accuracy of the model calculations, temperature sensors were embedded in core samples at K73 + 260 on the upper and outer lanes of Chongqing Inner Ring Expressway (G65-Chongqing-Guizhou Expressway). Twenty-four temperature sensors were embedded in nine core samples at three locations. Temperature data were collected from the pavement structure in real time. The pavement had a 4 cm thick SMA-13 upper layer and a 6 cm thick SMA-20 lower layer of asphalt horseshoe grease mixture, all of which were coarse. The base was a 34 cm thick layer of cement-stabilized macadam with 4% cement. The subbase was a 20 cm layer of graded macadam. The pavement structure and buried sensor locations are shown in Figure 3. The burial of an actual sensor in the field is shown in Figure 4.

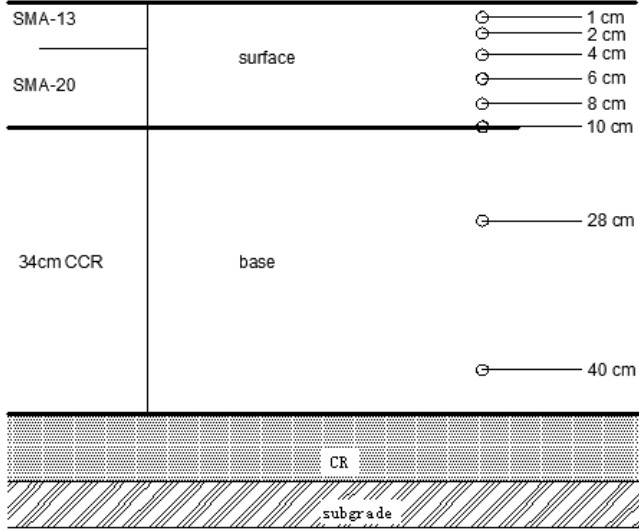

**Figure 3.** Pavement structure and sensor embedding diagram.

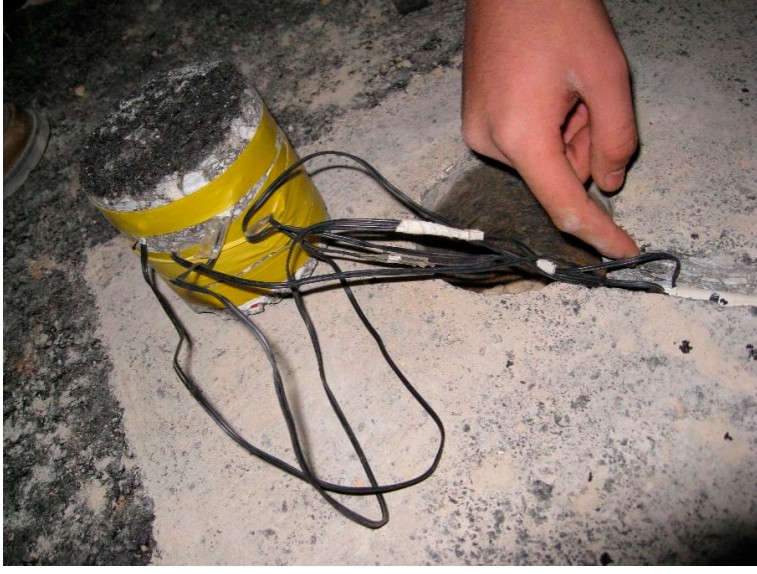

**Figure 4.** Burial of temperature sensor in the field.

The expressway had been open to traffic for less than two years. The pavement structure was deep and the surface was rough. The solar radiation absorptivity of the surface was 0.87. The values of the thermophysical parameters for the pavement materials are shown in Table 4.

**Table 4.** Thermophysical parameters calculated for the materials in the case study.

| Materials | $c$, J/(kg·°C) | $\rho$, kg/m$^3$ | $\lambda$, W/(m·°C) |
|---|---|---|---|
| surface | 984 | 2350 | 1.28 |
| base | 1084 | 2200 | 1.31 |
| subbase | 1069 | 2000 | 1.51 |

According to the numerical calculation model and boundary condition difference equation, the temperature in an asphalt pavement can be solved using Matlab. The parameters required as inputs by the program comprise two types: meteorological elements, i.e., the atmospheric temperature and solar radiation; and the thermophysical parameters of the material, i.e., the solar radiation absorptivity, thermal conductivity, specific heat, and density.

*4.2. Daily Variation of Asphalt Pavement Structure Temperature*

We considered an asphalt pavement structure in the Banan District of Chongqing as an example during 17–19 March 2017. The meteorological conditions are shown in Table 5. The calculated and measured temperatures in the pavement structure (2 cm) are shown in Figure 5.

**Table 5.** Meteorological conditions.

| Date | Maximum Temperature (°C) | Minimum Temperature (°C) | Total Solar Radiation (MJ/m$^2$) | Sunshine Hours (h) | Average Wind Speed (m/s) |
|---|---|---|---|---|---|
| 2017-03-17 | 24.9 | 16.8 | 6.731 | 1.8 | 0.4 |
| 2017-03-18 | 25.3 | 17.2 | 7.538 | 2.1 | 0.2 |
| 2017-03-19 | 25.7 | 17.7 | 7.316 | 1.9 | 0.3 |

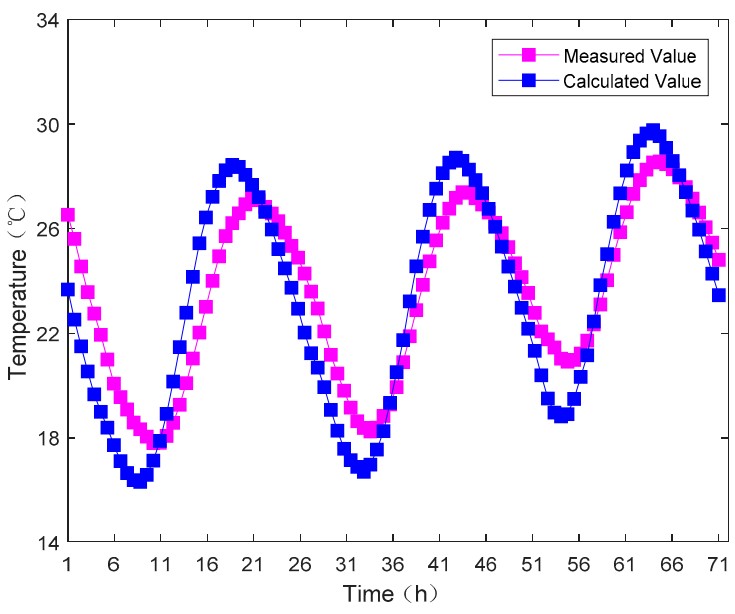

**Figure 5.** Comparison of the temperature in the asphalt pavement.

Figure 5 shows that the change in the temperature in the pavement structure exhibited a sinusoidal trend for several days, which was consistent with the changes in the air temperature. As the temperature

changed, the maximum and minimum temperatures in the asphalt pavement structure increased or decreased by various amounts. Compared with the measured data, we found that the numerical calculation model obtained high accuracy, where the difference was basically within 3 °C (the maximum difference occurred at 39 h and the value was 2.8 °C). In addition, the numerical calculation of the lowest temperature in the pavement structure was lower than the measured value, whereas the numerical calculation of the highest temperature was higher than the measured value. Therefore, the numerical calculation model obtained good accuracy and the results were within a safe range.

In particular, on 17 March 2017, the meteorological conditions used in the model were: maximum atmospheric temperature = 24.9 °C, minimum atmospheric temperature = 16.8 °C, total solar radiation = 6.731 MJ/m$^2$, sunshine hours = 1.8 h, and average daily wind speed = 0.4 m/s. According to the meteorological data, the total sunshine hours on this day were less than 2 h, so it was basically a rainy day. The temperature of the asphalt pavement was calculated at 6:00 am, and 12:00 pm, using the calculation model, and the results are shown in Figures 6 and 7.

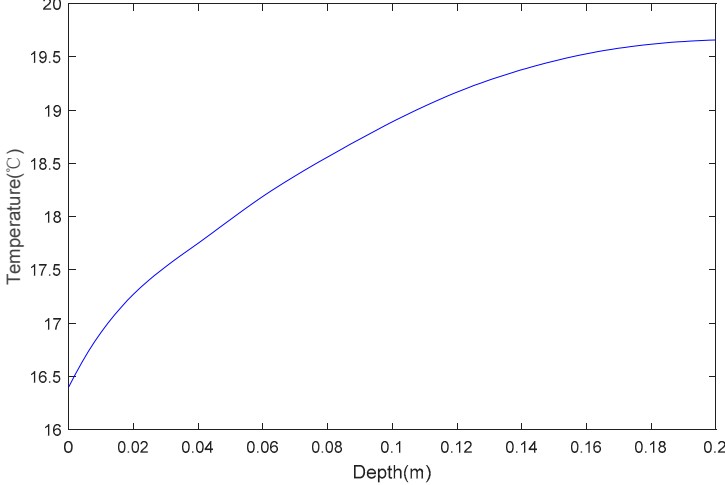

**Figure 6.** Temperature calculated for the asphalt pavement at 6:00 a.m.

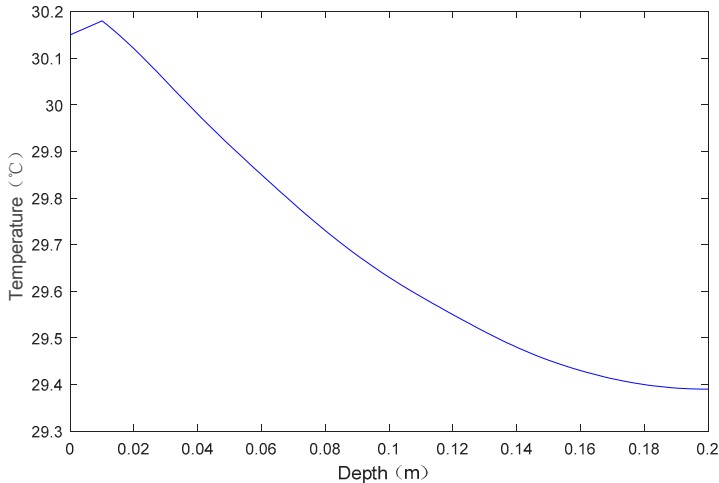

**Figure 7.** Temperature calculated for the asphalt pavement at 12:00 p.m.

At 6:00 am, the atmospheric temperature was low at 17.3 °C. The temperature of the pavement structure was also low at this time. Figure 6 shows that the surface of the road had the lowest temperature in the pavement structure at about 16.5 °C, which was slightly lower than the atmospheric temperature. The temperature of the pavement structure increased gradually in the depth direction and it was close to 20 °C at a depth of 0.2 m.

Figure 7 shows that at 12:00 pm, the temperature of the pavement increased greatly. The temperature of the pavement structure decreased in the depth direction. However, the highest temperature was not at the surface of the road, but instead it occurred about 1 cm below the surface of the road, probably due to the effect of the wind on the road surface. According to Figures 6 and 7, as the atmospheric temperature increased, the temperature in the pavement structure also increased, but the changes differed. The range of the temperature change under the road surface was about 5 cm above 10 °C, whereas the range of the temperature increase was lower in the depth direction. Thus, during the day, the layer about 5 cm below the road surface exhibited the maximum range for the temperature change, and this is also the upper layer thickness range in a high grade highway. This result provided a theoretical basis for the layered construction of a high grade highway. Thus, the upper layer should comprise high quality materials with better performance and greater resistance to temperature changes in order to improve the performance, but without increasing the cost excessively.

### 4.3. Temperature Gradient in the Asphalt Pavement Structure

In addition to studying the diurnal variations in temperature at a certain depth (2 cm) in the asphalt pavement structure, the temperature gradients were analyzed at different depths in the asphalt pavement structure, as shown in Figures 8 and 9.

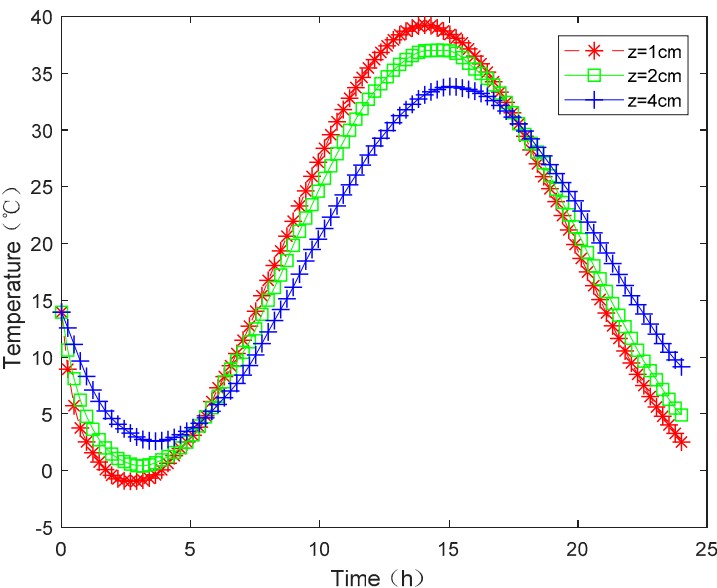

**Figure 8.** Calculated temperature in the asphalt pavement structure.

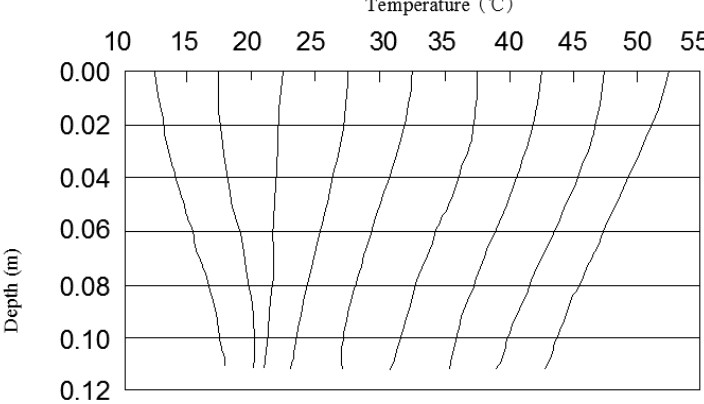

**Figure 9.** Temperature distribution in the asphalt pavement structure.

The meteorological conditions on 21 January 2018 used in the model were: maximum atmospheric temperature = 17.2 °C, minimum atmospheric temperature = 3.7 °C, total solar radiation = 11.351 MJ/m$^2$, sunshine hours = 7.6 h, and average daily wind speed = 0.3 m/s. Figure 8 shows the temperature in the asphalt pavement at different depths (0–4 cm) during a single day. The changes in the temperature at different depths in the pavement were similar to the changes in the air temperature, where the lowest value occurred at 5:00–7:00 a.m. and the highest at 1:00–3:00 p.m. However, as the depth increased, the times of the lowest and highest temperatures were delayed, with significant lags.

In terms of range of the temperature change, the surface of the asphalt pavement exhibited a large range during one day, where it reached 20 °C. The maximum temperature range appeared at 1 cm below the surface, where it approached 30 °C. This large difference in the range indicates a requirement for temperature-sensitive surface materials.

Figure 9 shows the temperature distribution at different depths in the asphalt pavement structure on 26 July 2018. The meteorological conditions on this day were: maximum atmospheric temperature = 40.2 °C, minimum atmospheric temperature = 21.7 °C, total solar radiation = 22.638 MJ/m$^2$, sunshine hours = 10.6 h, and average daily wind speed = 0.2 m/s. Due to the effects of high temperature and strong radiation, the high temperature range of 50–55 °C occurred on the road's surface. When the surface temperature was 52.5 °C, the temperature gradient was larger in the pavement structure, where it reached 9.7 °C. The other temperature curves moved to the middle of the calculated interval and the temperature gradient tended to decrease gradually. When the surface temperature was lower than 25 °C, the temperature in the pavement structure increased slowly with the depth. When the surface temperature was higher than 25 °C, the temperature in the pavement structure tended to decrease as the depth increased, and it gradually stabilized at about 30 °C (ground temperature).

## 5. Conclusions

- In this study, according to the basic principle of heat transfer, we established difference equations for various nodes using the finite difference method and a two-dimensional unsteady numerical model for calculating the temperature in an asphalt concrete pavement structure.
- Comparisons of the numerical results and the measured data for several consecutive days indicated that the numerical results maintained high accuracy, where the maximum temperature deviation was within 3 °C, and the numerical calculations of the lowest and highest temperatures were within a safe range. Thus, the method is suitable for practical applications in engineering.
- The pavement structure temperature and atmospheric temperature were highly correlated, where the relative changes were basically the same. However, as the depth increased, the times of onset for the high and low temperatures in the pavement structure were delayed, with obvious hysteresis.
- During a single day, the temperature changed greatly in the asphalt concrete surface, where the temperature difference exceeded 20 °C. This finding provides theoretical support to facilitate the improved design of asphalt surface materials and the selection of asphalt materials.
- In the depth direction, there was a temperature gradient of 10 °C in the pavement structure, but it differed from the temperature difference of 40 °C in the pavement surface during a single day. The temperature of the pavement structure was basically stable at about 30 °C in the 20 cm layer below the pavement, where it gradually approached the ground temperature.
- The thermophysical parameters are constant for asphalt pavement materials, but the thermophysical parameters of asphalt pavement materials change with the temperature and other factors in practice. Therefore, the effects of changes in the thermophysical parameters on the temperature in an asphalt pavement structure need to be investigated further in future research.

**Author Contributions:** Conceptualization, G.W.; Data curation, N.Z.; Methodology, B.C.; Software, C.C.; Supervision, G.W.; Validation, N.Z.; Writing-original draft, N.Z.; Writing-review & editing, B.C.

**Funding:** This research was funded by NFSC Project (Grant No. 51608085), and the funds of Science and Technology Project of Chongqing Education Committee (Grant No. KJ1740469).

**Conflicts of Interest:** The authors declare no conflict of interest.

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
