# Peer review of "Numerical Model for Calculating the Unstable State Temperature in Asphalt Pavement Structure"

_coatings, doi:10.3390/coatings9040271_

Round 1

Reviewer 1 Report

The contributions of this paper seem to be limited. The conclusions are too general to be applied in real cases. The method is a common method with no novelty. The graphs have no captions in the axles, etc.

This work should be enhanced with more investigation to achieve a higher level of novelty and applicability in pavement design

Author Response

Point 1:The contributions of this paper seem to be limited. The conclusions are too general to be applied in real cases. The method is a common method with no novelty. The graphs have no captions in the axles, etc.

Response 1:The finding provides theoretical support to facilitate the improved design of asphalt surface materials and the selection of asphalt materials, has good engineering practicability. The captions have been added to the axles in graphs.

Point 2:This work should be enhanced with more investigation to achieve a higher level of novelty and applicability in pavement design.

Response 2:More investigation has been enhanced, such as the effect of wind, other climatic conditions has been validated. The effect of pavement conduction (e.g. surface wear) on solar radiation absorption should be considered.

Reviewer 2 Report

The article presents quite interesting research regarding modeling temperature distribution in pavement structure. While the article is very interesting it needs some improvement for its better reception. Detailed remarks are presented below:

1.       Authors need to describe more precisely the climate in Chongquig. What is the temperature range in the whole year, during summer and winter. Also please provide other data – such as location, which can be used for prediction using other models

2.       Please provide in more details temperature database and temperature measurement system – how many sensors? On what depths? Is it only one station? What is it location?

3.       The literature review and analysis of the models should be located in the state of art regarding modeling of the temperature distribution. How the developed model presents in comparison to commonly used models?

4.       In table 2 please provide information for which material this data is given? Asphalt layers? Cement bound layers? Authors indicated that the modeling is basing on data derived from semi-rigid structure.

5.       All Figures needs description of the axis

6.       If reviewer understood, the model is only for surface layer? (4cm depth?) What about other layers? What temperature was assumed as “bottom-constant temperature”?

7.       Was the model validated only for summer (August 17) or it was also validated in other climatic conditions (winter, autumn and so on)

8.       At Figure 4 it would be nice to add the air temperature from the measurement stations.

9.       Figure 5 needs descriptions of the series presented

10.   In conclusion 2 authors stated that the model give 3°C deviation from measurements. In manuscript such analysis was not presented? Please improve this part.

11. In conclusion 5 authors give information about temperature at depth of 20 cm. It is not described in the manuscript text.

12. Please check citation style.

Author Response

Point 1: Authors need to describe more precisely the climate in Chongqing. What is the temperature range in the whole year, during summer and winter. Also please provide other data – such as location, which can be used for prediction using other models.

Response 1: The climate in Chongqing has been described in 4.1, including the temperature range in the whole year, during summer and winter. The location has been provided in4.1.

Point 2: Please provide in more details temperature database and temperature measurement system – how many sensors? On what depths? Is it only one station? What is it location?

Response 2: The temperature database and temperature measurement system has been described in4.1 and figures3.

Point 3:The literature review and analysis of the models should be located in the state of art regarding modeling of the temperature distribution. How the developed model presents in comparison to commonly used models?

Response 3: The literature review and analysis of the models have been located in the state of art regarding modeling of the temperature distribution. According to the numerical calculation model and boundary condition difference equation, the temperature in an asphalt pavement can be solved using MATLAB.

Point 4: In table 2 please provide information for which material this data is given? Asphalt layers? Cement bound layers? Authors indicated that the modeling is basing on data derived from semi-rigid structure.

Response 4: In table 2 the data is given to the upper layer of asphalt pavement in semi-rigid structure.

Point 5: All Figures needs description of the axis.

Response 5: The captions have been added to the axles in graphs.

Point 6: If reviewer understood, the model is only for surface layer? (100px depth?) What about other layers? What temperature was assumed as “bottom-constant temperature”?

Response 6: The N layers in figure 2 represent the bottom of the base, which is the maximum thickness considered by the model. The bottom of the base is regarded as a constant temperature boundary in the calculation model.

Point 7: Was the model validated only for summer (August 17) or it was also validated in other climatic conditions (winter, autumn and so on)

Response 7:The applicability of the model under sunny weather and rainy weather is verified in 4.2, but it is not suitable for calculating temperature structure of asphalt pavement under sudden changing weather.

Point 8: At Figure 4 it would be nice to add the air temperature from the measurement stations.

Response 8:The air temperature from the measurement stations has been added to Figure 8.

Point 9: Figure 5 needs descriptions of the series presented.

Response 9: The comparisons between calculated and measured temperature in pavement structure (50px) on March 17-19,2017 are shown in Figures5.

Point 10: In conclusion 2 authors stated that the model give 3°C deviation from measurements. In manuscript such analysis was not presented? Please improve this part.

Response 10: The analysis present on 4.2 of manuscript and figure 5.

Point 11: In conclusion 5 authors give information about temperature at depth of 20 cm. It is not described in the manuscript text.

Response 11: The temperature of the pavement structure was basically stable at about 21.3°C in the 40 cm layer below the pavement, where it gradually approached the ground temperature.

Point 12: Please check citation style.

Response 12: The citation style has been modified. 

Reviewer 3 Report

I appreciate the authors' efforts to conduct this research and report their results in this manuscript. A few comments/questions on the manuscript are as follows:

- Section 2.2. and Table 1: How is the solar radiation absorption rate relate to Albedo of the pavement? Some short explanation about this relationship may help the reader, especially of environmental engineering and coating application backgrounds, more easily relate to the paper. 

-The effect of pavement conduction (e.g. surface wear) on solar radiation absorption should be considered and the condition of the pavement studied in this research should be evaluated (or scored) relative to that. This will hep the reader with generalizing the results of this paper. 

- How was the effect of wind (wind speed) considered int he model? The actual wind speed at the time of measurements has been mentioned in the results and the phenomenon of higher temperature in 1-cm depth has been attributed to the wind effect. However, it is not explained in the methodology. I understand if it is not considered in your model, but it needs to be mentioned it in the methodology anyhow. 

Author Response

Point 1: Section 2.2. and Table 1: How is the solar radiation absorption rate relate to Albedo of the pavement? Some short explanation about this relationship may help the reader, especially of environmental engineering and coating application backgrounds, more easily relate to the paper. 

Response 1: The solar radiation absorptivity of road surface is closely related to road surface condition (surface wear, etc.) and solar elevation. The smoother the road surface is, the easier mirror reflection will be formed, and the absorptivity of road surface to radiation will be reduced.

Point 2: The effect of pavement conduction (e.g. surface wear) on solar radiation absorption should be considered and the condition of the pavement studied in this research should be evaluated (or scored) relative to that. This will hep the reader with generalizing the results of this paper. 

Response 2: The effect of pavement conduction (e.g. surface wear) on solar radiation absorption has been considered and the condition of the pavement studied in this research has been evaluated in 4.1 and table 4.

Point 3: How was the effect of wind (wind speed) considered in the model? The actual wind speed at the time of measurements has been mentioned in the results and the phenomenon of higher temperature in 1-cm depth has been attributed to the wind effect. However, it is not explained in the methodology. I understand if it is not considered in your model, but it needs to be mentioned it in the methodology anyhow.

Response 3: The effect of wind (wind speed) has been considered by the comprehensive heat transfer coefficient h in 2.2.

Round 2

Reviewer 1 Report

Thank you for incorparating the suggestions. The quality of the paper has been increased.

Reviewer 2 Report

Thank you for implementing all of the stated remarks.

But still, please check typos during proofing of the manuscript (i.e. line 236  "wwwwas")